# Influence of Manufacturing Parameters on Mechanical Properties of Porous Materials by Selective Laser Sintering

**DOI:** 10.3390/ma12060871

**Published:** 2019-03-15

**Authors:** Dan Ioan Stoia, Emanoil Linul, Liviu Marsavina

**Affiliations:** Department of Mechanics and Strength of Materials, Politehnica University of Timisoara, 1 Mihai Viteazu Avenue, 300 222 Timisoara, Romania; dan.stoia@upt.ro (D.I.S.); emanoil.linul@upt.ro (E.L.)

**Keywords:** porous materials, additive manufacturing, selective laser sintering, process parameters, sample orientation, tensile test, mechanical properties, energy absorption

## Abstract

This paper presents a study on the tensile properties of Alumide and polyamide PA2200 standard samples produced by Additive manufacturing (AM) based on selective laser sintering (SLS). Because of the orthogonal trajectories of the laser beam during exposure, different orientations of the samples may lead to different mechanical properties. In order to reveal this process issue, four orientations of the samples in building envelope were investigated. For data reliability, all the other process parameters were constant for each material and every orientation. The tensile tests highlight small differences in elastic properties of the two materials, while significant differences in strength properties and energy absorption were observed. Nevertheless, Young modulus indicates high stiffness of the Alumide comparing to PA2200 samples. The stereo microscopy reveals a brittle fracture site for Alumide and a ductile fracture with longitudinal splitting zones for PA2200. From the orientation point of view, similar properties of samples oriented at 0 and 90 degrees for all investigated mechanical properties were observed. However, tensile strength was less influenced by the sample orientations.

## 1. Introduction

In recent years, due to their exceptional properties, high/medium/low-density porous materials (PMs) with polymeric [1,2,3], ceramic [4,5,6] and metallic [7,8,9,10] matrix, have been widely used in engineering applications. The main mechanical and physical characteristics of PMs (materials with cells, cavities, channels or interstices) vary depending on the pore size, topology, and shape of the pores, as well as the porosity and composition of the solid material [11,12,13].

The additive technologies became widespread in all industries due to geometrical versatility of the produced samples [14,15,16]. Independently on the type of technology used, the accuracy and mechanical properties of the samples represent a challenging research issue [17,18]. The inconsistencies in shape, size, porosity and mechanical properties are associated with a series of technological parameters that are specific to each technology [19,20,21,22]. In order to establish a relation between technological parameters and real properties of the samples, investigations, such as mechanical testing (tensile, compression, bending), geometry and tolerances evaluation and microstructure of integer and fractured samples, have to be conducted [23,24,25]. Using lithography 3D printing, some authors obtained porous biocompatible materials for scaffolds [26].

Using a laser sintering system Mengqi et al. [27] analyzed the influence of orthogonal orientation planes on the resolution of wedges and lithospheres geometries. Polyamide 12 was used as building material for obtaining optically translucent samples. The resolution of the samples was acquired by stereomicroscopy for both XY and XZ working planes. The authors evidenced that building in vertical planes (XZ and YZ) significantly increased the contrast and resolution of the samples comparing to horizontal (XY) plane.

In order to identify the geometrical characteristics of additive manufactured samples, Guido et al. [28] used three additive processes: laser sintering, laser melting, and fused deposition modeling. The material used in the laser sintering process was polyamide PA2200. By designing basic geometric elements, the authors established the most suitable orientations, directions, thicknesses and radius according to the element. The samples evaluation was done by computing the dimensional variations after the geometric measurements. 

The technological parameters in the selective laser sintering (SLS) process generate a great influence on the mechanical properties of the resulting samples. Pilipovic et al. [29] highlighted the main flexural properties of finished products, according to some manufacturing parameters. They focus on the influence of the energy density used in the sintering process. Different energy densities were obtained by using different beam overlays while maintaining the power and the speed constant. Not only were the flexural properties determined but also the influence of the beam overlay on the density (mass) of the samples. 

In order to determine the mechanical properties of scaffold structures that can be used in medical engineering, Singh et al. [30] used polyamide PA2200 material in the SLS process. They constructed a geometric open porous structure by CAD modeling and built it using different energy densities. The mechanical strength was computed based on compressive testing. The values obtained were spread in direct relation to energy density. Based on variance analysis, the authors established that laser power, scan spacing, and layer thickness have more contribution to samples’ strength than scan speed.

Mousa [31] used the Taguchi method to investigate the influence of the SLS process parameters on the curling phenomenon of the samples. The material used in the study was polyamide 12 and PA 12 mixed with a rigid multiphase-coated particle. The purpose of the composite material was to modify the shrinkage and wrapping behavior during the cooling process. The conclusion of the study was that the greatest influence on the sample curling is associated with powder base thickness, as large as 90%. Beside this, other parameters, such as sample bed temperature, the filler ratio, and laser power, play a marginal role.

In order to characterize the composite PA-Al_2_O_3_, Berti et al. [32] conducted some mechanical tests on SLS samples. They built up a series of samples in different orientation angles, according to the transversal axis of the machine. The mechanical testing results reveal evident anisotropy in the growing direction and seem to be less sensitive to the sintering x and y directions. In addition, the effect of sintering direction becomes more evident at higher temperatures.

Goodridge et al. [33] propose an interconnected study on the factors that influence the SLS of the polymers. It seems that no currently available system solves the inconsistencies in dimensions and properties caused by the uneven temperature distribution during the process. Other identified factors of influence were energy density, sample orientation and placement, layer thickness, cooling rate, powder degradation, and reuse.

The dimensional performance of the samples obtained by two fused deposition modeling (FDM) technologies was investigated by Galantucci et al [34]. The correlation between the technological factors and the orthogonal dimensions of the samples was established. According to their work, the differences between the real and nominal specimen dimensions were in the range of 2 to 3 tenths of millimeters. The measurements were conducted using a digital microscope. 

Dimensional precision, flatness error and surface texture in FDM were the main concerns of Nuñez et al. [35]. The authors established the exact tolerance and surface finish that the FDM process can offer. The curing contractions lead to poor dimensional control of the sample, the deviations being in the order of hundreds of millimeters for a certain layer thickness. Decreasing the layer thickness to minimum, the deviations grow ten times, in the order of tenths of millimeters.

In order to study the effects of different laser exposure strategies on tensile properties of the samples produced by direct metal laser sintering technology Palumbo [36] they used an Al-based alloy produced by EOS (AlSi10Mg). The obtained results were evaluated using nested ANOVA statistics that point out the possibility of adjusting the laser exposure parameters as long as the energy density delivered in the powder layer is optimal. 

Most of the above-presented studies were focused on geometrical characterization than on the mechanical properties of polyamide materials, while other works focus on the mechanical properties but using different energy density to build the samples. Therefore, the purpose of this study was to investigate the main tensile mechanical properties of two different materials (PA2200 and Alumide), that were used to manufacture standard testing samples. Both materials were processed using the same set of technological parameters but at different orientation of the sample according to the building envelope, as we will explain in the following sections. In addition, this investigation presents the mechanical properties based on tensile testing, while other works used bending and compression tests. 

## 2. Materials and Methods 

### 2.1. Materials

Two commercially available powders developed by Electro Optical Systems-EOS GmbH, Krailling, Germany were used in the study: polyamide PA2200 and Alumide.

Based on polyamide PA12, the PA2200 powder is a multipurpose material that exhibits high strength and stiffness, good chemical and long-term stability, high detail resolution, biocompatibility according to EN ISO 10993-1 [37] and USP/level VI/121 °C and food contact approval in compliance with the EU Plastics Directive 2002/72/EC [38,39,40].

The general and thermal properties of PA2200 highly influence the manufacturing process and mechanical properties. These properties are: average grain size 56 µm according to ISO 13320-11 [41], bulk density of 0.45 g/cm³ according to EN ISO 60 [42], melting point 172–180 °C according to EN ISO 11357-1 [43], Vicat softening temperature B/50 according to EN ISO 306 [44] is 163 °C and using A/50 method 181 °C. 

The Alumide® is a product developed by homogeneous mechanical mixing of fine polyamide 12 (PA12) and aluminum particles. The sintered Alumide has porous/granular aspect to touch and visual inspection, a metallic appearance and higher stiffness than polyamide. It can be easily machined by milling, lathing or grinding, and it also can be polished and coated. Other advantages of Alumide confronted with PA2200 are: good geometrical precision, good density–rigidity balance, better thermal conductivity and acceptable bending strength [45,46,47,48]. The general and thermal properties are: average grain size 60 µm according to ISO 13320-11 [41], bulk density according to DIN 53466 [49] is 0.64 ± 0.04 g/cm³, melting point 172–180 °C according to EN ISO 11357-1 [43], Vicat softening temperature B/50 according to EN ISO 306 [44] is 169 °C, heat conductivity 0.5–0.8 W/mK measured at 170 °C using hot wire method. 

### 2.2. Methods

In order to accomplish the objectives of the study, the samples were manufactured, cleaned and painted in the middle section and then were subjected to mechanical testing.

The sinterization process was conducted on EOS Formiga P100 machine (EOS GmbH Electro Optical Systems, Krailling, Germany) that uses a 30 W CO_2_ laser to selectively sinter the powder particles. The machine preparation consists of setting up the building envelope and the powder barrels, setting up the process parameters and preparing the job file using the machine’s software. After the first building process, the machine was cleaned out by the polyamide powder and refilled with the Alumide powder in order to manufacture the second set of samples. During the post-processing time, the samples were cleaned by air blowing and painted in the middle section.

The EOS Formiga P100 machine consists of two main chambers were the temperature and oxygen content are controlled (Figure 1). The laser radiation is produced in a vertical laser tube, the beam being after conducted using rotating mirrors up to the scanning head. Here, deflecting mirrors ensure the deviation of the beam according to the focal plane. The focal distance is a hardware setting that assures a maximum energy density at the focal plane level. Three heating systems deliver the energy required for softening the powder, at temperatures in range 163–170 °C. The removal chamber is set at lower temperatures (150–153°C) for preventing the contractions that occur at cooling. 

The powder reaches the building chamber by means of a delivery system, and is uniformly spread on building plane by a sweeping blade. The geometry, surface roughness and coating of the sweeping blade are very important, any adhesion of the heated powder to the blade may represent a potential risk for the building process. 

In SLS additive manufacturing there is an important number of input parameters that significantly influence the geometry and mechanical properties of the outcoming samples. Some of the parameters are controllable (chamber temperatures, laser energy density, location, and orientation of the samples in the building envelope, layer thickness, scaling factors, materials, scanning strategies), while other factors, such as temperatures gradients in powder volume and at the surface, local apparent density of the powder, contractions at cooling, can be considered noise [19,31,40,50].

The controllable process parameters used for the entire samples are presented in Table 1. The power of the laser beam (P), its velocity (v) and scan spacing (d) are three parameters that build up the energy density (ED), required for sinterization of the powder bed (Equation (1)) [29,51]. The relation between these three parameters is graphically presented in Figure 2.
(1)$$ED=\frac{P}{v\cdot d}\left[J/mm{}^{2}\right]$$

The temperatures inside building and removal chambers are generated by the radiation of the electrical heaters and provide the required energy of softening the polyamide powder and the reduction of temperature gradient on Z direction.

The geometrical characteristics of the laser beam can be observed in Figure 3. The diameter of the laser spot is Φ = 0.42 mm for Formiga P100 machine [38], however the curing zone is larger (Φ1) and dependent on the energy density (velocity and the power of the laser beam). A value of Φ1 = 0.68 mm was determined by lkgun for 9.4 W of power and 700 m/s beam velocity [52]. In order to obtain the nominal dimension of the sample, the laser beam trajectory in contour lines was reduced by an offset value (e) for compensating the radius of the beam spot. The scan spacing of the laser beams (d) was set to 0.25 mm. At this value, an overlapping of 28.5% of the laser spot area is achieved, leading to a good sinterization of powder lines. The scanning strategy used was alternant, meaning that one layer was sintered in X direction and the next layer in Y direction, obtaining in this way a crosshatching for the orthogonally oriented samples. The layer thickness (T) plays an important role in the energy density generated in the powder volume [53]. For polyamide, the recommended layer thickness [38] is 0.1–0.2 mm. In order to compensate the shrinkage at cooling, the samples were scaled (enlarged) uniformly by 2.2%. A different scaling factor for Z axis was not considered in this study due to the reduced height of the samples. 

The second part of the study comprises the mechanical testing of the samples. These were subjected to tensile testing on INSTRON 8800 machine. All samples were individually coded prior to testing. The tests were performed at room temperature up to the breaking point using 5 mm/min loading speed. After testing, the results of force, displacement, and longitudinal and transversal strains were further processed according to ISO 527-1:2012 [54].

Representative samples of both materials and each orientation were selected and prepared for metallographic evaluation that was conducted on a Kruss stereo microscope, having the purpose of identifying the fracture site and fabrication layers.

## 3. Results and Discussions

The geometrical model of the test sample was constructed in SolidWorks 2013 according to the dimensional recommendation of ISO 527-1:2012 [54]. The sample was then placed in the virtual environment of the machine using Materialise Magics 10.0 where different orientations were considered. The samples orientation angles (SOA) according to horizontal direction were 0°, 30°, 45°, 90° (Figure 4a). For each orientation angle, nine samples have been equally distributed in the building plane, keeping a safety distance of dx = dy = 5 mm of the physical envelope edges. Due to insufficient space in the building plane for placing all the samples at all the angles in one layer, these were divided into four-layer groups. Between each layer group, a distance of dz = 10 mm was used for preventing the temperature influence among groups. Because of the orthogonal trajectories of the laser beam during the hatching process and the orientations of the samples, three types of grids were obtained (Figure 4b). With respect to the longitudinal axis of the sample, 0° and 90° orientations lead to symmetrical orthogonal sinterization lines, while 30° and 45° lead to oblique sinterization lines that may generate different mechanical properties. 

After positioning all samples in the building envelope and verifying the mesh structure of the volumes, the file containing geometrical information was passed through a series of specific steps that included all the process parameters in it. The steps comprised layer by layer division of the volume, checking for geometrical errors in every layer, expansion of the contours according to the scaling factors and process parameters assignment. At the end of these steps, conducted on the machine software RP Tools and EOS PSW, the job file was created and ready for sending to Formiga P100. The building process starts by heating up the machine chambers to the values specified in Table 1. This process step takes about two hours and represents a mandatory stage for building. The actual building process time was 6 h and 20 min and was elapsing almost linearly, since the total sintering area remains constant in every layer. Two images showing the directions of hatching are presented in Figure 5.

When the building process was finished, the machine cooled down for 12 h. All samples were post-processed by air blowing and coded according to the orientation, resulting in 36 samples of Alumide and another 36 of PA2200. All of them were weighed using a digital scale and measured using a digital caliper. 

The mechanical tensile tests were conducted on INSTRON 8800 machine according to ISO 527-1:2012 [54]. Prior to fixing, all samples were painted in black on the middle section in order to reach a good contrast for the reflective dot markers. The reflective dots were placed in the middle section of the samples and represent the gage length and width (Figure 6). The video extensometer allows measuring the relative axial and transversal displacement of the middle section of the sample. The tensile test was conducted up to a sudden load drop identifying the event of a fracture, using a loading speed of 5 mm/min. The room temperature during tests was 20 °C.

The individual stress (σ)-strain (ε) curves of PA2200 and Alumide samples with four different angle orientations (0°, 30°, 45°, and 90°) are shown in Figure 7. The σ–ε curves are similar for each group of samples (PA2200 and Alumide) in the elastic zone, but the stress and strain at break are different for each sample. Due to the difficulty in analyzing a graph that contains 36 curves, in Figure 7 and Figure 8 only the most representative curves (1 curve/orientation) are presented [55,56,57].

From these representative tensile curves, it is possible to observe that the initial behavior of the PA2200 and Alumide samples is linear. After this linear-elastic behavior, all investigated samples show a gradual increase in stress during the elongation without showing a specific yield point. There is a smooth transition from elastic-to-plastic deformation region, and then the stresses decreased drastically at an almost constant value of the strain.

For a better interpretation of the results, the experimental stress-strain curves of the PA2200 samples were compared with Alumide samples, having the same orientation angle. As shown in Figure 8 and Table 2, the elastic properties of Alumide samples are higher than those of PA2200. However, the strength properties and energy absorption capacity of the PA2200 are much higher than those obtained in the case of Alumide samples. The PA2200 samples fail between 12–16% of tensile strain, under maximum stress of 33 MPa, while Alumide samples fail between 5–6% of tensile strain, under maximum stress of about 15–17 MPa.

Table 2 presents the main mechanical properties (Young’s modulus–E, Quasi-elastic limit–σ_e_, Yield strength–σ_y_, Tensile/maximum strength–σ_m_, Elongation at break–ε_b_, Poisson ratio–ν and Energy absorption–W) of the PA2200 and Alumide tensile tests performed at room temperature with different angle orientations. 

Clearly, it can be deduced from the σ-ε curves and Table 2 that the reinforcement has completely changed the behavior and the main mechanical properties of the investigated PMs. To emphasize the influence of the reinforcements, the corresponding mean values of the tensile properties are plotted in Figure 9. In addition, in order to better quantify this effect, the main properties (E, σ_e_, σ_m_, ε_b_, ν and W), were investigated as a function of sample angle orientation.

From Figure 9 and Table 2, it is noticeable that the elastic properties and Poisson ratio are not strongly affected by the type of powders used in the study. Very little effect of both SOA and powder type on Poisson ratio is observed. However, PA2200 has higher ν values of up to 6–17% (according to SOA) compared to Alumide. The absolute variation of the Poisson ratio (Figure 9e) is less than 0.9% for Alumide and between 1.6–2.0% for PA2200 and can be considered as constant within this range of sample orientation angle. The only higher property of Alumide compared to PA2200 is the Young modulus. This elastic property together with the low value of the elongation at break, also known as fracture strain, indicates increased stiffness of the Alumide samples. The Young modulus increases slightly at angles of 0 and 90° compared to 30 and 45°, as observed in Figure 9a.

Based to their graphical representation (Figure 9), all other material properties such as σ_e_, σ_m_ and ε_b_ are quite similar and are thus discussed together. Figure 9 shows that the σ_e_, σ_m,_ and ε_b_ dependence on the sample orientation angle is opposite to that of Young’s modulus. The quasi-elastic limit (Figure 9b) and tensile strength (Figure 9c) of the PA2200 samples are between 1.66 and 2.25 times (according to SOA) higher than the Alumide ones. On the other hand, the elongation at break (Figure 9d) shows values up to 2.59 times higher for PA2200 compared to Alumide.

The energy absorption as a function of sample orientation angle is plotted in Figure 9f. It can be seen that the W at break of PA2200 is at least 5.5 times (82%) higher than the W of Alumide. This percentage difference is visible and constant for all investigated SOA.

Due to the differences in melting temperatures and heat transfer of aluminum and polyamide powders, the sinterization occurs between polyamide particles only. In Alumide, the presence of aluminum particles restricts the possibilities of polyamide particles bounding, this phenomenon leads to lower mechanical strength and lower absorbed energy comparing to pure polyamide. On the other hand, the stiffness of aluminum particles increases the Young modulus of Alumide. 

Subsequently to mechanical testing, eight samples corresponding to the four orientations and the two materials were selected. The stereo microscopy examination at 20 X magnification reveals the orientation layers and the fracture sites for all selected samples (see Figure 10). 

While for PA2200 material the fabrication layers are visible and easy to identify, for Alumide samples these are harder to be identified, due to the shining effect of aluminum particles. The PA2200 samples manifest a ductile fracture presenting longitudinal splitting zones, while the Alumide samples reveal a more brittle fracture section.

## 4. Conclusions

The main mechanical properties, according to four different sample orientation angles (0°, 30°, 45° and 90°) and the two materials (PA2200 and Alumide) used in selective laser sintering additive manufacturing are investigated in the paper. The experimental results were obtained using the process parameters indicated in Table 1. 

It was observed that the type of powders does not have a considerable influence on the elastic properties (Young’s modulus and Quasi-elastic limit) and Poisson ratio. The only higher property of Alumide compared to PA2200 is the Young modulus. This elastic property together with the low value of the elongation at break indicates increased stiffness of the Alumide samples. The energy absorption at break of PA2200 is at least 82% higher than the energy absorption of Alumide. This percentage difference is visible and constant for all investigated sample orientation angles.

Fracture site microscopy reveals sinterization layers for PA2200, and less visible layers for Alumide. The PA2200 samples manifest ductile fracture while Alumide manifests brittle fracture.

## Figures and Tables

**Figure 1 materials-12-00871-f001:**
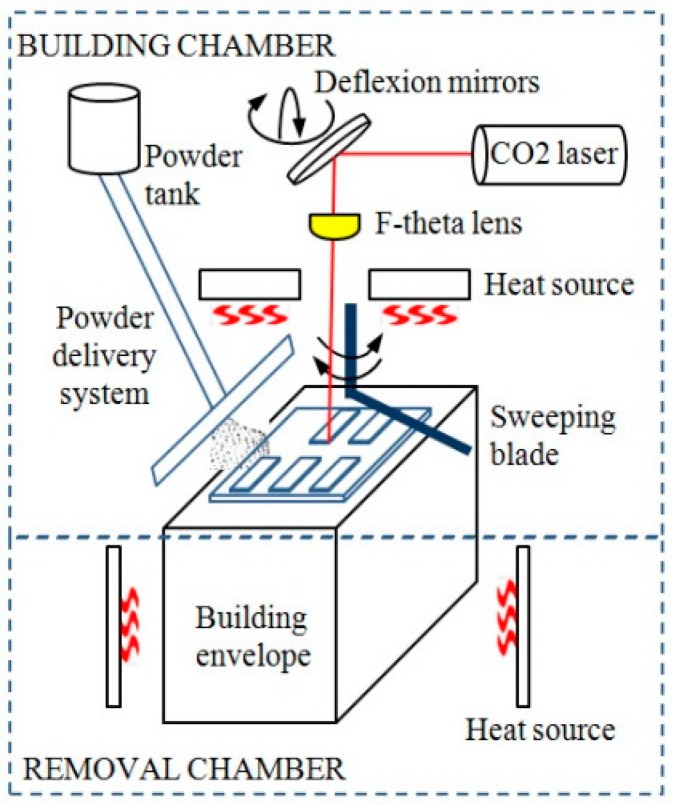
Structure of EOS Formiga P100 selective laser sintering (SLS) machine.

**Figure 2 materials-12-00871-f002:**
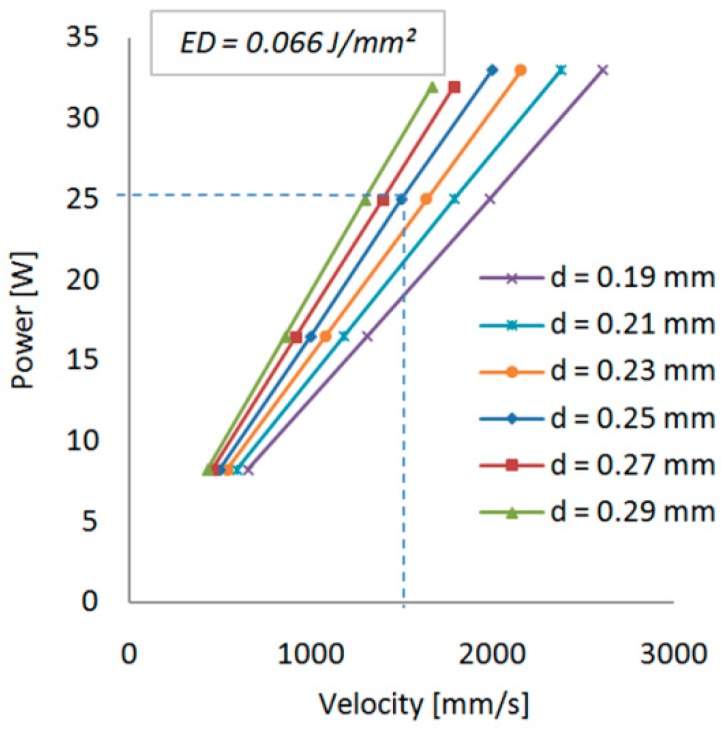
Linear dependency of parameters that create the energy density.

**Figure 3 materials-12-00871-f003:**
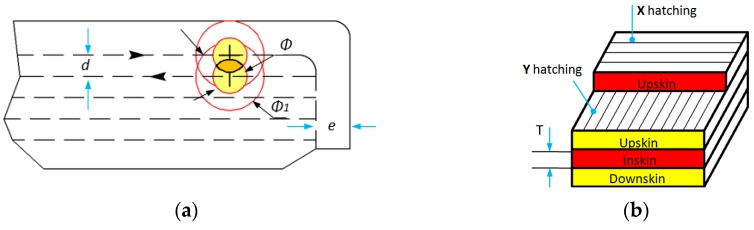
Geometrical aspects of laser spot and building layers. (**a**) laser trajectories and overlapping of the focal spot; (**b**) scanning strategies and layer types.

**Figure 4 materials-12-00871-f004:**
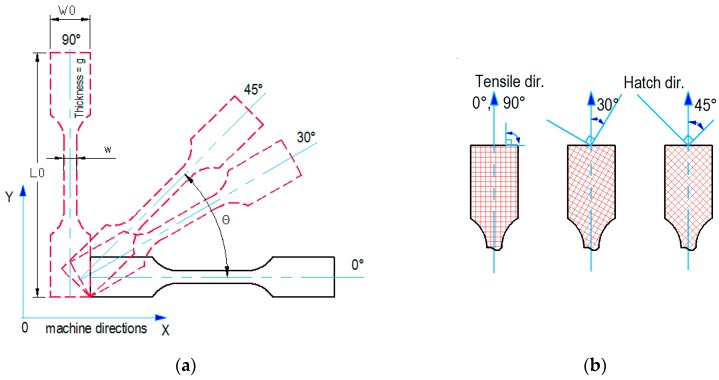
Test sample model. (**a**) orientation of the sample inside the building envelope; (**b**) hatching lines according to the longitudinal axis (tensile direction) of the sample.

**Figure 5 materials-12-00871-f005:**
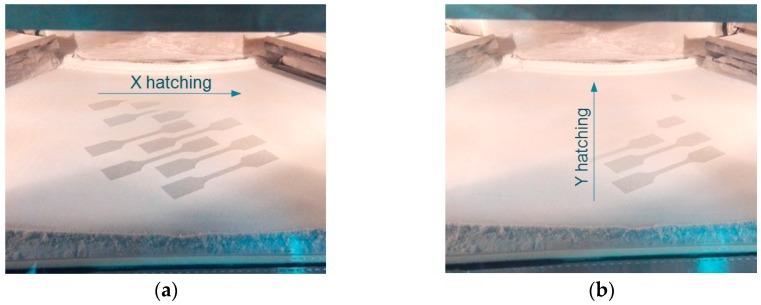
Images from sinterization process for 45° orientation of the samples. (**a**) hatching of layer *k* in X direction; (**b**) hatching of layer *k+1* in Y direction of the machine.

**Figure 6 materials-12-00871-f006:**
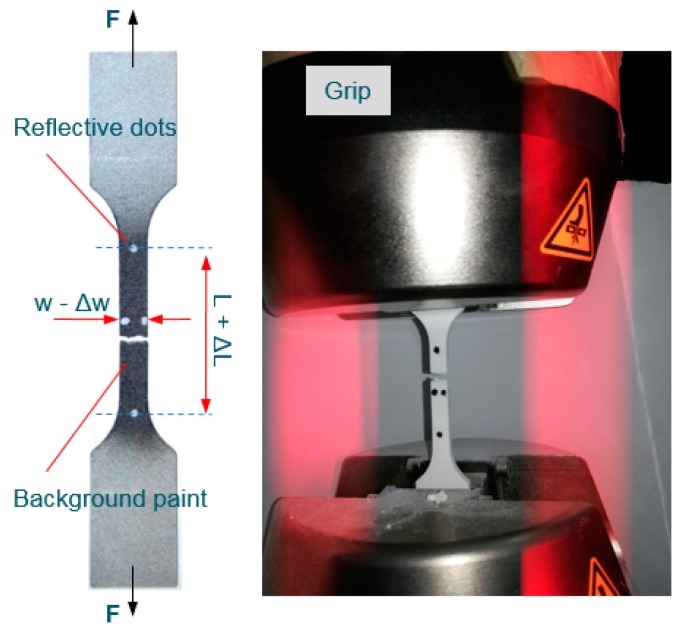
Alumide sample after tensile testing.

**Figure 7 materials-12-00871-f007:**
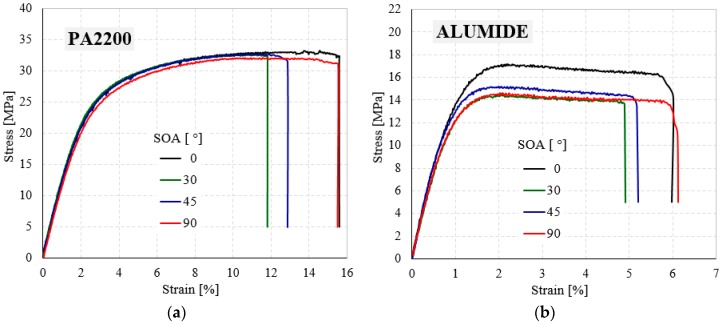
Typical tensile engineering stress - engineering strain curves of PA2200 (**a**) and Alumide (**b**) samples for different angle orientations (0°, 30°, 45°, and 90°).

**Figure 8 materials-12-00871-f008:**
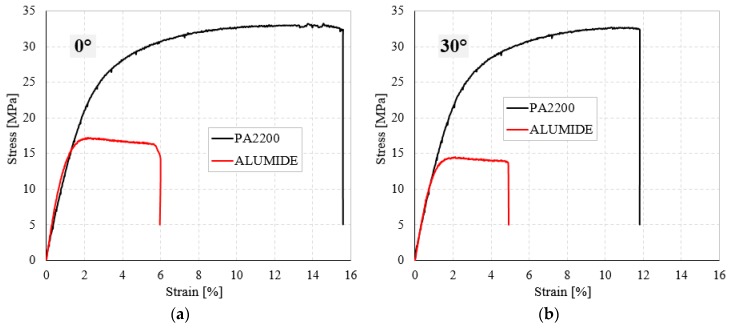
The experimental tensile stress-strain curves for PA2200 and Alumide samples at different orientation angle: (**a**) 0°; (**b**) 30°; (**c**) 45° and (**d**) 90°.

**Figure 9 materials-12-00871-f009:**
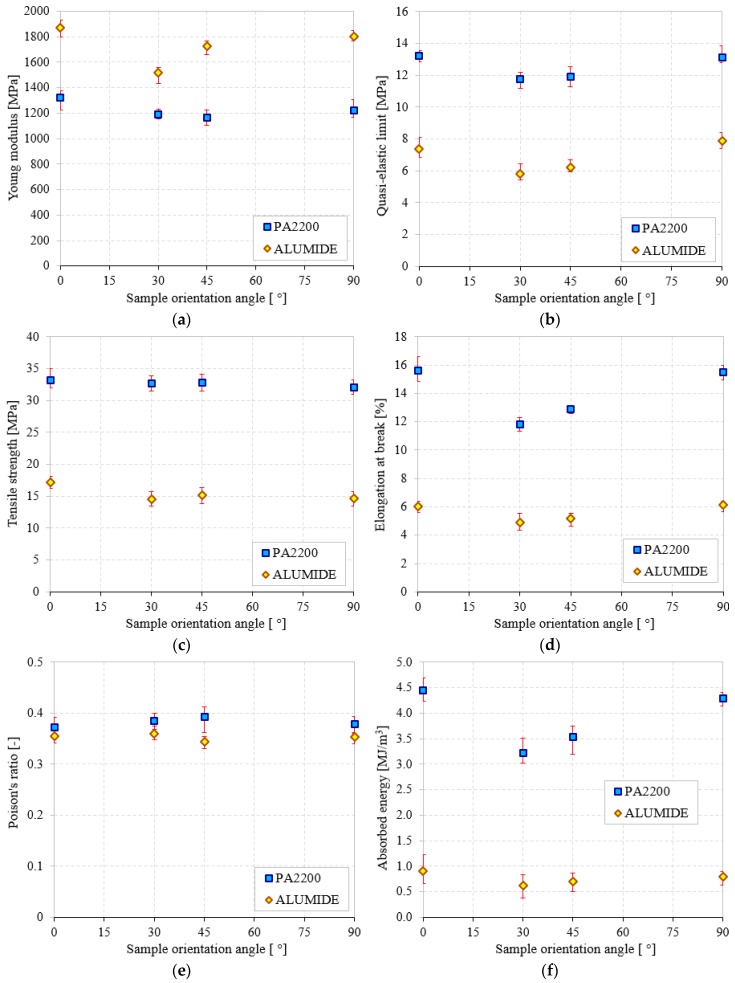
The main tensile mechanical properties of 3D printed PMs (PA2200 and Alumide) according to sample orientation angle (0°, 30°, 45°, and 90°): (**a**) Young modulus; (**b**) quasi-elastic limit; (**c**) tensile strength; (**d**) elongation at break; (**e**) Poisson’s ratio; (**f**) absorbed energy.

**Figure 10 materials-12-00871-f010:**
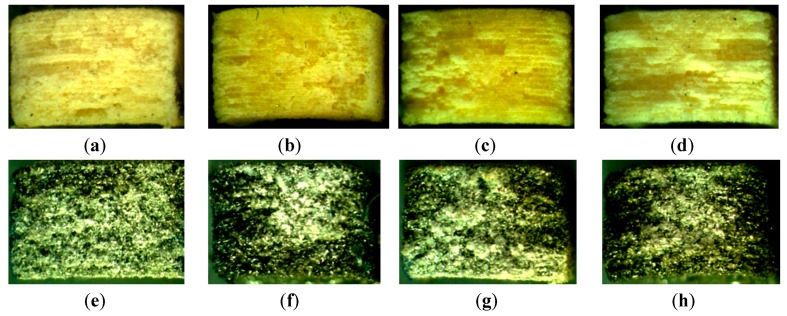
Fracture section of representative samples: PA2200-0° (**a**); PA2200-30° (**b**); PA2200-45° (**c**); PA2200-90° (**d**); Alumide-0° (**e**); Alumide-30° (**f**); Alumide-45° (**g**); Alumide-90° (**h**).

**Table 1 materials-12-00871-t001:** Process parameters used in laser sintering of the samples.

Laser Power (W)	Laser Velocity (mm/s)	Scan Spacing (mm)	Energy Density (J/mm^2^)	Beam Offset (mm)	Building Chamber Temp. (°C)	Removal Chamber Temp. (°C)	Layer Thickness (mm)	Scaling Factors (%)
25	1500	0.25	0.066	0.15	170	153	0.1	2.2

**Table 2 materials-12-00871-t002:** Tensile mechanical properties of investigated PA2200 and Alumide porous materials (PMs).

Material	α (°)	E (MPa)	σ_e_ (MPa)	σ_y_ (MPa)	σ_m_ (MPa)	ε_b_ (%)	W^1^ (MJ/m^3^)	ν (-)
PA2200	0	1321.90 ± 73.1	13.21 ± 0.36	21.12 ±0.73	33.24 ± 1.55	15.62 ± 0.91	4.45 ± 0.23	0.42 ± 0.016
	30	1187.00 ± 36.5	11.75 ± 0.50	20.42 ± 0.68	32.69 ± 1.24	11.82 ± 0.56	3.21 ± 0.25	0.39 ± 0.013
	45	1162.50 ± 58.2	11.89 ± 0.63	20.61 ± 0.56	32.81 ± 1.33	12.87 ± 0.20	3.54 ± 0.28	0.42 ± 0.025
	90	1224.10 ± 69.3	13.11 ± 0.53	20.52 ± 1.02	32.08 ± 1.14	15.52 ± 0.51	4.29 ± 0.14	0.40 ± 0.018
Alumide	0	1871.00 ± 66.4	7.39 ± 0.62	13.16 ± 0.57	17.17 ± 0.99	6.01 ± 0.42	0.90 ± 0.29	0.35 ± 0.012
	30	1518.10 ± 63.0	5.81 ± 0.50	11.78 ± 0.86	14.49 ± 1.15	4.91 ± 0.60	0.62 ± 0.23	0.36 ± 0.014
	45	1722.70 ± 53.7	6.23 ± 0.39	11.82 ± 0.75	15.16 ± 1.27	5.20 ± 0.48	0.69 ± 0.19	0.34 ± 0.012
	90	1799.00 ± 41.2	7.89 ± 0.52	12.35 ± 0.38	14.65 ± 1.15	6.12 ± 0.35	0.79 ± 0.14	0.35 ± 0.017

Note: 1: Energy absorption values corresponding to stress at break (σ_b_).

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
