# Peer review of "Influence of Manufacturing Parameters on Mechanical Properties of Porous Materials by Selective Laser Sintering"

_materials, 2019, doi:10.3390/ma12060871_

Round 1
Reviewer 1 Report
At this point there is no suggestions or comments for the authors. The paper is well written, although the result of the paper is not as promising as was expected.
Author Response
# Reviewer 1
At this point there is no suggestions or comments for the authors. The paper is well written, although the result of the paper is not as promising as was expected.
Answer: The authors are grateful and appreciate the positive evaluation of the manuscript.
Reviewer 2 Report
The manuscript stated methods to investigate the influence of various orientation planes and two powder materials (Alumide and PA2200) on the mechanical properties of SLS-manufactured samples.
-A large part of "3. Results and Discussions" were dedicated to describing methods and protocols. In the section, the results are described in detail. However, there is a lack of data analysis and discussion beyond simply comparing the mechanical properties between the two materials.
-Figures need more specific descriptions.
Author Response
We want to thank the reviewers for their valuable inputs to our paper. They indeed helped us to further improve the manuscript. In the following, the changes of this revision are detailed together with the referee’s comments:
List of responses to the reviewer(s) comments
# Reviewer 2
The manuscript stated methods to investigate the influence of various orientation planes and two powder materials (Alumide and PA2200) on the mechanical properties of SLS- manufactured samples.
-A large part of "3. Results and Discussions" were dedicated to describing methods and protocols. In the section, the results are described in detail. However, there is a lack of data analysis and discussion beyond simply comparing the mechanical properties between the two materials.
Answer: According to the reviewer suggestion, the main part of the section 3 was moved to the section 2. Some additional discussions were added at the end of the “Results and discussion” section. All average data together with the standard deviations were presented in the Table 2.
-Figures need more specific descriptions.
Answer: Some specific descriptions were added to Figures 7 and 8, highlighted in the manuscript. In adition, all the results from Figure 9 presents the error bars. Figure 6 has been replaced in order to reduce its brightness.
Reviewer 3 Report
Please find my suggestions from the reviewing process.
1. The authors show a good lit review mentioning what their studies have shown regarding similar materials and processes. The motivation needs a bit more reinforcement though as they mentioned that, “Guido et al [24] studied PA 2200, also, Singh et al. [26] used polyamide PA2200 material in SLS process”. Could the authors emphasize what their manuscript repots differently from these previous studies where the same materials and processes have been investigated?
2. Numerical values should be reported with deviations, scatter plots should have error bars.
3. The following observations are recommendations only, left to the discretion of authors to be included or not.
Line 39, Analyzed
Line 72, conducted
Line 141, preparing metallographic samples for further observations using…
Line 144 – 171. looks more of a process description, shouldn’t this be in section 2?
Line 155, may represent a potential, instead of “being”
Line 178, is there any evidence/reference of Phi1 value?
Line 206, “series of steps” Which ones?
Line 207, including the process parameters instead of “that implement all the process parameters in it”
Line 210, this process step instead of “this process”
Line 221, prior fixing
Line 225, up to a sudden load drop identifying the event of fracture…
Line 228, seems to be redundant information.
Line 232, “final failure”, please explain
Line 235, are these examples instead? or why these are representative? Why not all curves are included? Please comment clearly
Line 255-256, have a significant influence instead of “completely changed”
Some figures show excesive brightness, (Fig 6 for example) adding too much brightness with the background, would be desirable the figure to be softened
I would advice to get the title modified so can mention either process or materials, at the moment is too generic and may be a bit misleading.
Author Response
We want to thank the reviewers for their valuable inputs to our paper. They indeed helped us to further improve the manuscript. In the following, the changes of this revision are detailed together with the referee’s comments:
List of responses to the reviewer(s) comments
# Reviewer 3
Please find my suggestions from the reviewing process.
1. The authors show a good lit review mentioning what their studies have shown regarding similar materials and processes. The motivation needs a bit more reinforcement though as they mentioned that, “Guido et al [24] studied PA 2200, also, Singh et al. [26] used polyamide PA2200 material in SLS process”. Could the authors emphasize what their manuscript repots differently from these previous studies where the same materials and processes have been investigated?
Answer: In the last paragraph of the introduction section (highlighted), the authors added some additional explanations regarding the novelties of this paper.
2. Numerical values should be reported with deviations; scatter plots should have error bars.
Answer: The standard deviations were added to numerical values from Table 2. Also, error bars were added to Figure 9.
3. The following observations are recommendations only, left to the discretion of authors to be included or not.
-Line 39, Analyzed
-Line 72, conducted
-Line 141, preparing metallographic samples for further observations using…
Answer: The mentioned sentences were corrected and highlighted in the manuscript. “Representative samples of both materials and each orientation were selected and prepared for metallographic evaluation that was conducted on Kruss stereo microscope, having the purpose of identifying the fracture site and fabrication layers”
-Line 144 – 171. looks more of a process description, shouldn’t this be in section 2?
Answer: The mentioned paragraphs were moved to the section 2.
-Line 155, may represent a potential, instead of “being”
Answer: The mentioned sentences were corrected and highlighted in the manuscript.
-Line 178, is there any evidence/reference of Phi1 value?
Answer: Information added: “…however, the curing zone is larger (Φ1) and dependent by the energy density (velocity and the power of the laser beam). A value of Φ1 = 0.68 mm was determined by Özkan İlkgun for 9.4 W of power and 700 m/s beam velocity [52]”.
-Line 206, “series of steps” Which ones?
Answer: Information added: “The steps comprise layer by layer division of the volume, checking for geometrical errors in every layer, expansion of the contours according to the scaling factors and process parameters assignment.”
-Line 207, including the process parameters instead of “that implement all the process parameters in it”
-Line 210, this process step instead of “this process”
-Line 221, prior fixing
-Line 225, up to a sudden load drop identifying the event of fracture…
Answer: The mentioned sentences were corrected and highlighted in the manuscript.
-Line 228, seems to be redundant information.
Answer: The redundant information was removed.
-Line 232, “final failure”, please explain
Answer: The individual stress – strain curves of PA2200 and Alumide samples with four different angle orientations are similar in the elastic zone.
-Line 235, are these examples instead? or why these are representative? Why not all curves are included? Please comment clearly
Answer: For each orientation (0, 30, 45 and 90 deg) 9 samples were manufactured and tested. In the Figures 7 and 8 are presented only the most representative curves (1 curve/orientation) because it was very difficult to analyze a graph with 36 curves. Additionally, all data were averaged and standard deviation was computed and clearly presented in the Table 2.
-Line 255-256, have a significant influence instead of “completely changed”
Answer: The mentioned sentence was corrected and highlighted in the manuscript.
-Some figures show excessive brightness, (Fig 6 for example) adding too much brightness with the background, would be desirable the figure to be softened
Answer: Figure 6 has been modified.
I would advise to get the title modified so can mention either process or materials, at the moment is too generic and may be a bit misleading.
Answer: Title has been modified: Influence of Manufacturing Parameters on Mechanical Properties of Porous Materials by Selective Laser Sintering
Round 2
